# Evolution towards Smart and Software-Defined Internet of Things

**Muhammad Aneeq Abid [1], Naokhaiz Afaqui [1], Muazzam A. Khan [2,3,\*], Muhammad Waseem Akhtar [1], Asad Waqar Malik [1], Arslan Munir [4], Jawad Ahmad [5] and Balawal Shabir [1]**

1  School of Electrical Engineering and Computer Science (SEECS), National University of Sciences and Technology (NUST), H-12, Islamabad 44000, Pakistan; 11beemabid@seecs.edu.pk (M.A.A.); nafaqui.msit18seecs@seecs.edu.pk (N.A.); engr.waseemakhtar@seecs.edu.pk (M.W.A.); asad.malik@seecs.edu.pk (A.W.M.); bshabir.dphd18seecs@seecs.edu.pk (B.S.)
2  Department of Computer Science, Quaid-i-Azam University, Islamabad 44000, Pakistan
3  Pakistan Academy of Sciences, G-5, Islamabad 44000, Pakistan
4  Department of Computer Science, Kansas State University, Manhattan, KS 66506, USA; amunir@ksu.edu
5  School of Computing, Edinburgh Napier University, Edinburgh EH10 5DT, UK; j.ahmad@napier.ac.uk
\*  Correspondence: muazzam.khattak@qau.edu.pk

**Abstract:** The Internet of Things (IoT) is a mesh network of interconnected objects with unique identifiers that can transmit data and communicate with one another without the need for human intervention. The IoT has brought the future closer to us. It has opened up new and vast domains for connecting not only people, but also all kinds of simple objects and phenomena all around us. With billions of heterogeneous devices connected to the Internet, the network architecture must evolve to accommodate the expected increase in data generation while also improving the security and efficiency of connectivity. Traditional IoT architectures are primitive and incapable of extending functionality and productivity to the IoT infrastructure's desired levels. Software-Defined Networking (SDN) and virtualization are two promising technologies for cost-effectively handling the scale and versatility required for IoT. In this paper, we discussed traditional IoT networks and the need for SDN and Network Function Virtualization (NFV), followed by an analysis of SDN and NFV solutions for implementing IoT in various ways.

**Keywords:** Internet of Things (IoT); Software-Defined Networking (SDN); Network Function Virtualization (NFV); massive connectivity; industrial IoT; IoT evolution

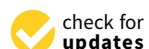


## 1. Introduction

The modern era is characterized by the combination of two fundamentally different concepts: "Internet" and "Things". Despite their vast differences, the two have managed to coexist in a network, which has been termed the Internet of Things (IoT). In addition to people communicating with machines and each other, IoT has enabled machines to communicate with each other [1–3]. Machines are now connected and interact with one another independently, resulting in an increase in data in a mesh network [4–6].

IoT is growing at an exponential rate with technological advancements enabling more and more devices, or "things", to act smart and connect on the network. Applications, networks, and sensors make up the three-layered architecture of the IoT [7,8]. Users can interact with the diligent computing of IoT through the application layer. The network layer, as the name implies, is in charge of network operations and data forwarding, while the sensor layer is in charge of data collection [9,10].

IoT has changed our approach to work, from our day-to-day personal tasks in our micro lives to business models in the economy and around the world. IoT applications can be found in all aspects of life, from society to industry and the environment. Due to contextual sensing, the number of application domains is constantly growing. In terms

of technology and use cases, the IoT has undergone a continuous cycle of evolution and upgrade. As a result, we have seen IoT architectures evolve from their traditional state to highly efficient, smart, and software-defined systems with virtualization.

Mobile and personalized IoT-based healthcare services can be realized by using personal computing devices and mobile internet access [11]. From precision agriculture to food production, processing, storage, distribution, and consumption, IoT in the Food Supply Chain (FSC) can cover the entire farm-to-plate process. Physical objects can be monitored in real-time from origin to destination using IoT in transportation, improving efficiency, reliability, safety, and quality of goods delivered. In the mining field, IoT can help workers avoid accidents. The firefighting organization can detect hazards in real-time and prevent fire incidents by implementing IoT in firefighting scenarios.

The capacity and capability of traditional architecture and networking techniques to cater to IoT are not promising. Hence, it gave rise to the concepts of Software Defined Networking (SDN) and Network Functions Virtualization (NFV). SDN is a network architecture that separates the control and data planes [12]. The primary objective is to separate and operate the infrastructure layer and the application layer while centralizing control plane functions. By decoupling the control plane from the data plane, SDN performs a vertical partition in the network layer of the IoT architecture, giving the structure the flexibility and adaptability it requires to execute the functions.

To comprehend the idea of NFV, we must first understand the concept of virtualization. Virtualization, as the term suggests, is the process of creating something logical (and virtual) from a physical source. The goal of virtualization is to separate the hardware from the software. NFV is the rationing of available bandwidth and resources into channels that can be used and allocated independently by creating virtual networks within the same network and architecture as a result of this phenomenon. Virtualization divides the network layer's decoupled controls and data plane horizontally, separating network functions from physical functions.

Although SDN and NFV have different origins and come from different organizations, the two concepts can be combined to create a stronger, more diverse, and cost-effective architecture capable of handling the ever-increasing burden of IoT.

*Motivation*

IoT architectures have seen a rising trend of transformation and updates since their inception. The IoT ecosystem has been evolving to meet the model's requirements and expectations, as well as the growing demands of users. IoT has been molding continuously to address the challenges at first to meet the fundamental requirements for IoT, such as security and privacy [13,14]. IoT alone will not be able to meet the demands of today's world. Security is no longer a reactive phenomenon to be addressed when the need arises, but rather a proactive and by-design requirement in today's world [15].

Security is critical at both the network's edge and its core. In today's world, IoT must be dynamic, flexible, and distributed enough to ensure availability in the event of a disaster [16,17]. Furthermore, with the increasing amount of data generated by IoT devices, efficient and intelligent channel allocation methods and a multi-tiered control layer framework are required to avoid communication bottlenecks and ensure smooth data transfer [18]. IoT does not have the functions or capability to address these issues or meet the demands of today's world on its own. As a result, technologies such as SDN and NFV form the foundation of IoT architectures to create the stable, secure, efficient, smart, flexible, and intelligent systems that are expected and overcome traditional IoT architectural limitations such as weak security, data distribution inefficiency, provenance, and traceability of sources, excessive human intervention, lack of interoperability, and service-awareness in device configuration [19].

A lot of research has been carried out in these domains of IoT, SDN, and NFV, resulting in the evolved form of smart and efficient IoT frameworks we have the privilege of using today. However, in terms of the evolution of IoT architectures, from primitive and

traditional frameworks to highly evolved software-defined and virtualized platforms, there is a missing link. This research gap has necessitated the creation of this comprehensive survey, which captures the various stages of IoT structures and their advancements in an effort to better estimate future requirements and current weaknesses by studying the evolution cycle. Therefore, we cover nearly every aspect of IoT development in this paper, including IoT classification and applications, smart and cognitive IoT architecture, smart software-defined IoT, IoT architecture evolution, and state-of-the-art IoT simulators. The current state of IoT challenges and future directions are also discussed in this paper.

Figure 1 shows the taxonomy of the rest of the paper, which is organized as follows. In Section 2, IoT classification and applications are discussed. Section 3 explains the smart or cognitive IoT architecture, followed by Section 4, which describes smart software-defined IoT. The evolution of IoT architecture is given in Section 5. State-of-the-art IoT simulators are described in Section 6, followed by IoT challenges and future directions in Section 7, and finally, the paper is concluded in Section 8.

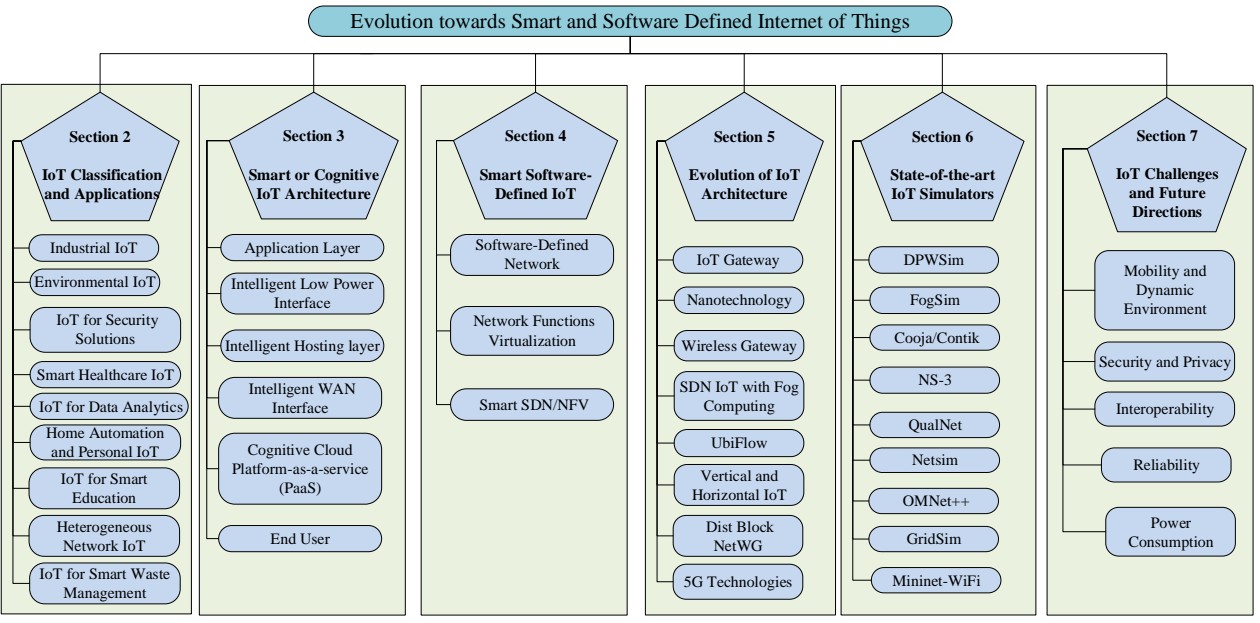

**Figure 1.** Taxonomy of the paper.

## 2. IoT Classification and Applications

IoT has changed our way of doing business, from our day-to-day personal tasks in our micro lives to business models in the economy and around the world. IoT applications can be found in all aspects of life, from society to industry to the environment. There are many different application domains for IoT applications, some of which will be discussed in the following sections. This section delves into the various classes and applications of IoT networks.

### 2.1. Industrial IoT

IoT has connected everything, including vehicles, to be able communicate with one another, enhancing the concept of smart cities and Vehicular Ad hoc Networks (VANET) [20,21]. Many areas are investigating the use of IoT in transportation systems to create more efficient and intelligent vehicular networks. Li Ping Qian et al. in [22] propose a scalable and long-term hybrid IoT framework that addresses efficient multiple access and computational offloading at the physical and MAC layers to support the efficient transmission, computation, and caching of IoT big data generated by the network's large number of vehicles. In another attempt, Lei Zhai et al. in [23] use the IoT to investigate the best resource allocation for pushing compute and storage resources to edge devices in

order to reduce average service response time. Similar attempts are being made to extend automotive-based IoT applications to the cloud to further enrich and enhance the VANET services [24].

IoT has impacted industrial standards as well, as it has other aspects of life circles [25]. IoT has applications in all industries, including agriculture, education, telecommunications, manufacturing, and medicine. Kai Fan et al. investigate the importance of IoT in the healthcare system and propose a lightweight authentication scheme for a cloud-based Radio-Frequency Identification (RFID) healthcare environment to address the privacy and security needs of the environment while maintaining cost-effectiveness [26]. Borja Martinez et al. examine the disconnect between actual IoT implementation in the manufacturing process and academic and theory [27]. Similarly, Monu Bhagat et al. examine the various applications of IoT in the agriculture industry, leading to smart farming to improve productivity, efficiency, and global market while reducing human intervention, cost, and time in their research [28].

### 2.2. Environmental IoT

The application of IoT can be extended to monitor and impact the environment as well. Environmental IoT can be applied to both indoor and outdoor environments. The first can fall into smoke detection, fire alarms, light detection, movement detection, indoor heat, and moisture monitoring, etc. [29], whereas the latter lies in the large outdoor deployments such as traffic monitoring; lightening and cloud detection; rain monitoring; pollution monitoring; and earthquake, chemical hazards, and flood detection; etc. [30]. Outdoor environmental IoT reliability and maintenance are difficult due to harsh climate conditions and high maintenance costs, whereas indoor environmental IoT reliability and maintenance are less challenging due to low cost and ease of being used [31].

It is observed that the open nature deployment of the IoT network and communication protocols for reliable operation is time-consuming and costly. There are energy management and environmental maintenance concerns for the smooth operation of open nature outdoor IoT deployments. However, modern techniques, such as machine learning, blockchain, and industrial automation, can be applied for the enhancement of reliability and cost reduction in environmental IoT. Such applications are studied in [32–35], where the authors propose a real-time, fine-grained, and power-efficient air quality monitor system based on aerial and ground sensing to keep a check on air pollution which is becoming the largest environmental health risk.

### 2.3. IoT for Security Solutions

Every industry requires security. With the advancement in edge technologies and multi-application usage, it is only evident that users demand seamless and secure access to their data and applications [36]. However, through traditional methods of securing edge and access, this user experience expectation cannot be met. It is through IoT that users can access their edge applications with the security they need in the consumer-friendly manner they desire. Aleksandr Ometov et al. in [36] propose a multi-factor authentication mechanism to address the security challenges discussed to grant reliable user-friendly access to advanced IoT applications. In a similar attempt to exploit IoT for security applications, Jianbing Ni et al. in [37] examine the architecture of mobile edge computing in an attempt to enhance security and efficiency in data usage.

Security and safety are some of the very important characteristics of IoT. IoT devices communicate with each other to provide sufficient security in an environment. IoT sensors sense the surroundings and inform about the current state of the environment. This helps to prevent any unwanted events in an environment [38]. For instance, in an autonomous car, a sensor is measuring the pressure of tires so it can help to prevent the tire bursting if pressure somehow crosses a threshold. In a heavy industrial area, IoT can prevent the damage of machines by measuring their temperature. Similarly, smart wearables can

monitor a harmful environment, and wireless body area networks (WBAN) can monitor the physiological parameters providing safety and security [39].

### 2.4. Smart Healthcare IoT

The healthcare industry's major goal is to deliver better healthcare to everyone, everywhere in the globe, at any time. This should be executed in a way that is more patient-friendly and cost-effective. Therefore, health monitoring devices need to be applied to increase the efficiency of patients' health care. Today's medical world faces two challenges in patient monitoring: first, the requirement for healthcare providers and caretakers to be present at the patient's bedside, and second, the patient's confinement to bed and connection to huge apparatus [40]. These requirements must be met to provide flexible and friendly patient care, and as bioinstrumentation and telecommunications technologies advance, it is becoming more feasible to design a home-based vital sign monitoring system that can gather, display, record, and transmit physiological data from a human body to any location.

### 2.5. IoT for Data Analytics

After the emergence of IoT, big data analytics experienced a boom. Millions of devices are continuously communicating with each other, and a large amount of data are accumulated. The data accumulated in this fashion are quite huge and can be used to perform data analytics. The machine learning and data mining techniques can be used to extract some useful features from the huge data [41]. The data are the fuel for today's digital transformation. Although the management of this huge data is a concern in IoT, the analysis of these data can be used for the healthy benefit of society. Analyzing big data can effectively be used for disaster management [42].

### 2.6. Home Automation and Personal IoT

IoT devices can be remotely managed without human intervention. This has affected the rise of the concept of smart cities, where all devices are communicating and behaving smartly to make the life of the common man simpler and easier. Different types of analytics can be performed on these data to find some relations. This helps to make the community take effective decisions based on these intelligent computations. These computations in IoT improved the living style of a common individual. The authors improved lives with better health management, better traffic handling, better weather predictions, smart homes, smart grids, etc. [43,44].

### 2.7. IoT for Smart Education

Students can now learn more effectively, efficiently, flexibly, and pleasantly thanks to the advancement of new technology. Researchers use smart devices to connect to digital resources via a wireless network and immerse themselves in individualized and seamless learning. Smart education, a term that emphasizes learning in the digital age, is gaining popularity. For smart learners who need to master 21st-century knowledge and skills, a four-tier framework of smart pedagogies and ten key qualities of smart learning environments are offered. Individual-based personalized learning, group-based collaborative learning, individual-based personalized learning, and mass-based generative learning are all part of the smart pedagogy paradigm [45].

### 2.8. Heterogeneous Network IoT

Heterogeneous networking is the revolution in the field of device connectivity. Heterogeneity is a very prominent feature that makes the machine-to-machine (M2M) communication possible with varying requirements of those machines. Different devices are working under different technologies and using different protocols. IoT brings all these diverse requirements under an umbrella and makes communication possible with these variants. The heterogeneous networking made billions of devices talk to each other in an

efficient way which gives rise to new horizons of interconnectivity including the Internet of vehicles (IoV) and the Internet of everything [46–48].

### 2.9. IoT for Smart Waste Management

Waste management models based on the IoT play a significant role in improving the quality of life and improving human well-being by increasing energy efficiency, improving governance, and lowering costs. Due to unprecedented population increase, increasing waste generation has become a significant challenge in developing countries.

Many issues have been investigated in the literature and found to have a strong correlation with the high waste material and hardships associated with managing it in a smart city, according to the findings. These problems are the result of insufficient collection and disposal mechanisms for waste material, an increasing trend in the urban population, and a lack of smart technology to assist in the operation of the municipal solid waste management system. As a result, due to the large amount of waste that has accumulated everywhere, waste management has become a difficult task.

Insung Hong et al. in [49] devise an IoT architecture for smart garbage waste management system aiming at increasing mobility of endpoints and enhancing communication all the while optimizing cost and energy by using resource-friendly protocols and sensors. The architecture addresses the IoT requirements of reliability, mobility, service continuity, user convenience, and energy efficiency. As a result, the operational costs are reduced as well as the overall food wastage. However, this model is faced with challenges. As the smart bins are mobile and battery-powered, enhancing the battery life while keeping it economical is a challenge for the system. Additionally, even though operational costs have been reduced by this model, unless the maintenance costs are minimized, the cost of the overall project is not going to be optimized.

## 3. Smart or Cognitive IoT Architecture

IoT makes it possible to connect the resource-constrained devices which continuously share information. The evolution of the IoT gave them the power to speak so that the devices can directly talk with each other with the help of protocols. These are different types of IoT devices that communicate with each other. They can be divided into low end, middle end, and high-end IoT devices [50]. The low end is resource-constrained which has small computing power, whereas the middle end is less resource-constrained as compared to the low end and provides more processing capabilities. High-end devices, on the other hand, are single-board computers (SBC) that have enough memory, storage, and computing power.

Figure 2 depicts a cognitive IoT network architecture. It can be seen that different types of communication technologies make this connectivity possible in IoT devices such as Wi-Fi, Bluetooth, Zigbee, laptop, scanner, base station, access points, and cloud network, etc. [51]. In the following sub-sections, we explain how these devices constitute a cognitive IoT network with the aid of AI/ML.

### 3.1. Application Layer

The application plan is in charge of providing the user with application-specific services. It defines several applications for the IoT, including smart homes, smart cities, and smart health. There are billions of IoT devices capable of sensing, processing, actuating, and communicating with the internet. They are communicating with each other through cellular networks or a gateway in the personal wireless local area network. Device-to-device (D2D) communication in an IoT environment is carried out through multiple application layer protocols and application programming interfaces. The communication protocols at the application layer are application-dependent and rely on multiple choices, such as security goals, data integrity, data confidentiality, and data availability.

### 3.2. Intelligent Low Power Interface

The intelligent low power plan is in charge of short-distance communication. Near Field Communication (NFC) is a technology that enables IoT devices to communicate within millimeters. By placing two NFC-enabled IoT devices in close vicinity, data can be transmitted in seconds. It employs RFID technology and magnetic field fluctuations to send data between two NFC-enabled devices. NFC employs the same 13.56 MHz frequency as high-frequency RFID in active and passive mode operation. In active mode, both devices create magnetic fields, while in passive mode one device generates the field while the other employs load modulation to send data. Batteries benefit from passive mode to save power. The proximity between devices is useful for safe transactions such as payments. Finally, unlike RFID, NFC allows for two-way communication. Consequently, practically every smartphone today has NFC [52].

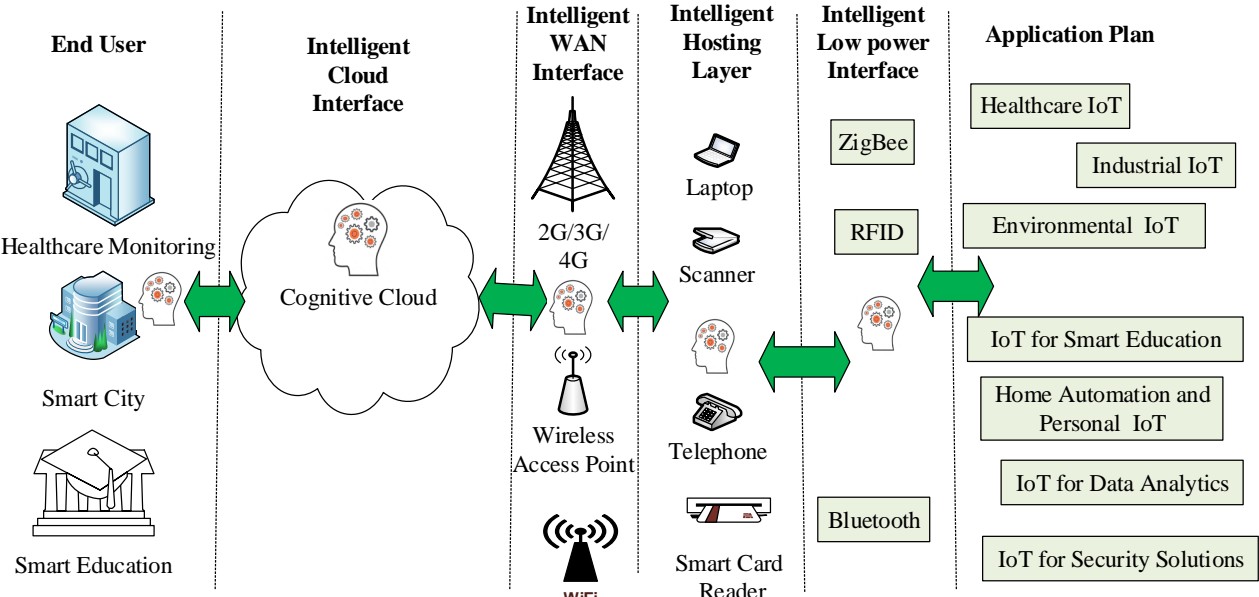

**Figure 2.** Cognitive IoT network architecture.

### 3.3. Intelligent Hosting Layer

The intelligent hosting layer provides various functions for the programmer, allowing the hardware details to be masked. This improves smart thing connectivity and makes it simpler to provide various services. The intelligent hosting layer plays an important role in achieving the requirements of a complete IoT architecture which include interoperability, scalability, multiplicity, security and privacy, abstraction provision, spontaneous interaction, and unfixed infrastructure [53].

Multiple variations in intelligent hosting have been highlighted through time to address the various requirements. A few prominent hosting layer devices are based on RFIDs and Wireless Sensor Networks, and other sensing platforms are Task Computing Framework (TCF), Triple based Distributed Middleware (TDM), Service Oriented Architecture (SOA), UBIWARE, TinyREST, Global Sensor Network (GSN), and Fosstrak, in addition to Robotic Operating System (ROS). Each of these hosting devices brings certain advantages to the table corresponding to specific domains of IoT ranging from simplicity, user-friendliness, and interoperability to mobility, efficiency, and manageability. Even though these platforms show promising capabilities, their outputs are limited and constrained. The challenges they face include resource management and efficiency, scalability, security, and configurability. AI and machine learning play an important role to overcome these challenges by assigning the resources intelligently and efficiently.

### 3.4. Intelligent WAN Interface

Non-IP technologies such as RFID, NFC, and Bluetooth have a limited range. Therefore, they can not be used in applications that require a vast region to be monitored by multiple sensor nodes spread over the area. A wireless sensor network (WSN) connects tens to thousands of sensor nodes wirelessly. They collect environmental data and send it to gateway devices that send it to the cloud via the Internet.

WSN nodes can communicate directly or via multihop. Sensor nodes are limited in power and computational resources, whereas gateway nodes are not. WSNs commonly use star, mesh, and hybrid network topologies for wide area networks (WAN). ML/AI at this interface enables the IoT network to intelligently select WAN protocols and networks as per the requirements of the user.

### 3.5. Cognitive Cloud Platform-as-a-Service (PaaS)

Traditional IoT architectures have been integrated with the cloud to obtain desired effects. F.TongKe in [54] proposes an IoT architecture with cloud computing to meet China's agricultural industry's technological priority and resource limits. Similarly, Jayavardhana Gubbi et al. proposed an IoT architecture based on cloud computing [55]. This architecture, that is a .NET (*pronounced as dot NET*)-based application and development PaaS, presents the integration of IoT elements such as sensors (RFIDs, wireless sensor networks (WSNs), addressing schemes, data storage, and analytics and visualization) through a cloud-centric approach. Challenges faced by this model include standardization of protocols and frequency bands, security and privacy of data, as well as a geographic information system(GIS)-based visualization and quality-of-service.

### 3.6. End User

The end-user of the cognitive IoT network could be the officials from the health care monitoring industry, e-education systems, monitoring systems in smart cities, robotics control, and monitoring industries.

## 4. Smart Software-Defined IoT

IoT has the sensing capabilities to sense different phenomena in the surroundings. These sensing capabilities make these devices monitor different activities to take the decision accordingly. The devices sense data from the surroundings and share these data with the millions of devices accordingly to develop a connected environment. For example, IoT sensors can sense the environment to measure the emission of carbon dioxide in the surroundings. It can help the community to make health-conscious decisions to make the environment human-friendly [56]. The sensed data are crucial for the decision making, and in case of sensed data loss, it is retrieved [57].

### 4.1. Software-Defined Network

The traditional architecture for IoT is struggling to cope with the requirements of the expanding technology. Conventional network structures are specialized in their functions. Their roles are pre-defined, and they are hard-coded for the job they are intended for. Due to resource constraints, it is not feasible to specify and feed multiple rules and complex roles for various dedicated tasks pre-hand [58]. This makes them less adaptive and devoid of the flexibility desired to shoulder the upcoming demands of IoT.

Software-Defined Networking brings the kind of control, flexibility, and adaptability that promises a stable IoT structure. Through the separation of the control plane from the data plane, SDN enables it to achieve its targets through traditional infrastructure. This approach is highly attractive as it is cumbersome, and in some situations not even feasible, to replace the entire structure for implementing new technologies. When the control plane operates independently, it offers a high degree of flexibility and eases to configure changes to the system and maintain adaptability and cater expansion.

Zhijing Qin et al. [59] proposed Mulinetwork Information Architecture (MINA) which is a layered SDN controller approach towards IoT that addresses the problem of heterogeneity of the system. It increases interoperability by enabling multi networking in Software-Defined Networking. MINA has a multilayer architecture, where the data collection component collects Network and Device information from IoT multi-network and stores it into databases used by layer components on the left. Layers provide abstraction as it hides the details of network devices which enables tasks completion flexibly.

The task-resource mapping layer decides which device and application should be used to complete the task. Lower level flow and network layers decide which networks should be used and how the application flows should be routed across the network. The applications of MINA extend to smart homes which are a cluster of heterogeneous devices and networks. MINA makes the system flexible and an efficient manager of tasks, flows, networks, and resources. The main disadvantage is that security and privacy issues are not explicitly addressed.

Yaser Jararweh et al. [60] proposed a software-defined system (SDsys) as a promising solution to mitigate the challenges of the traditional IoT and especially the data security. SDIoT uses SDN to abstract all the controls and management operations from the underlying devices. A complete package is included in SDsys covering software-defined network (SDN), security (SDsec), and storage (SDstore). The physical layer has a sensor cluster that contains all assets and devices as well as a database cluster that contains information about each sensor and also stores raw data from each sensor.

Middleware contains IoT controllers, SDN-C, SDStore-C, and SDSec-C. Westbound APIs are used to communicate with other controllers to scale. The Data-as-a-Service (DaaS) layer contains user applications that facilitate the access and interaction with stored data by the end-users using the Northbound APIs. A disadvantage is the energy-efficient scheduling and Quality-of-Service(QoS) per-flow features are not catered for in this solution.

### 4.2. Network Functions Virtualization

SDN has brought, to quite a large extent, the flexibility and scalability desired by IoT networks that were traditionally lacking. However, technology today is evolving at a rate faster than was possible before. The needs and desires of IoT architectures are changing accordingly. The networks demand to be stretched to even greater extents while enhancing security. To cater to these evolving requirements, SDN can be embedded with virtualization technologies to further strengthen and increase the parameters of the IoT networks, discussed so far, in addition to providing the functionalities of a multi-tenant environment, which was originally not feasible with SDN alone. In this section, we will study the NFV based SDN IoT architectures and analyze them accordingly.

Mike et al. in [10] explains that traditional architectures of IoT fail to meet the current requirements at various domains and, hence, need to be enhanced. SDN and NFV's integration to IoT brings about interoperability, discoverability, security, efficiency, management, and the desired scalability. Existing business models are infused with SDN and virtualization to see an increase in profitability trends.

### 4.3. Smart SDN/NFV

With the advent of 5th generation mobile technologies, problems and limitations faced by prior SDN and NFV based IoT models can be addressed with promising improvements. An SDN environment infused with NFV is shown in Figure 3 where the network function virtualization infrastructure (NFVI) layer is responsible to provide interoperability in a multivendor environment. J.O.Lucena et al. present a network slicing concept in 5G technology that can enhance the IoT architectures focusing on 5G applications [61]. The authors believe that network slicing can address the problem of accommodating multiple applications and users in a single channel to increase resource usage efficiency and cater to rising demands.

L.Tello et al. in [62] present a solution for advanced 5G connectivity in an SDN-based IoT system enhanced by virtualization. The solution presents a flexible and centralized platform with mobility management and scheduling. The load balancing is made mobility aware through virtualization, which also makes resource utilization efficient. However, the solution still needs reliable security.

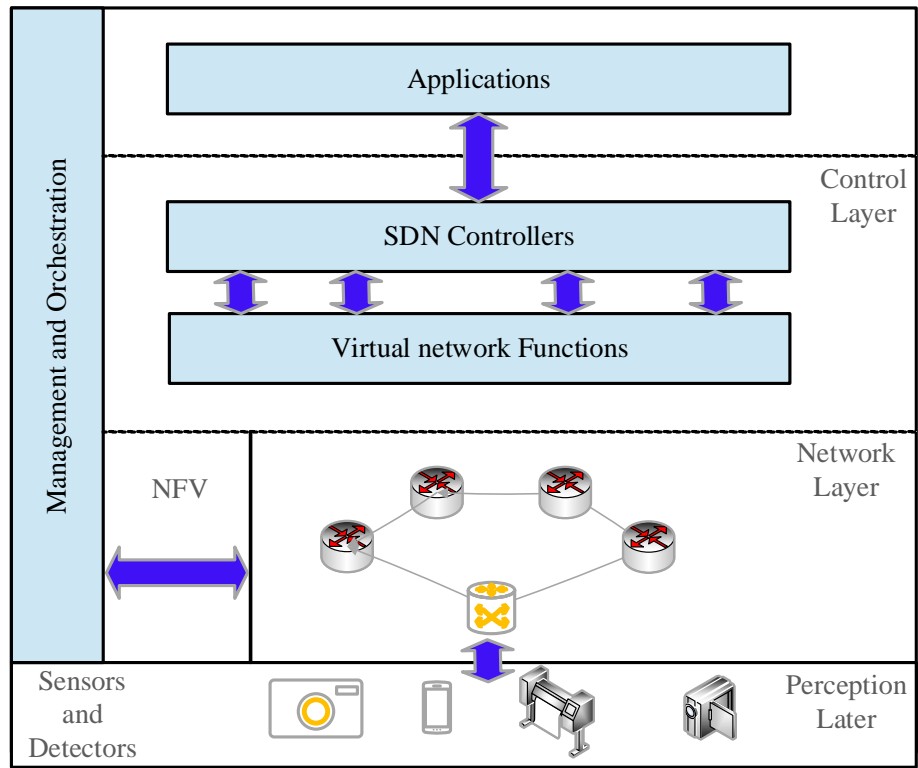

**Figure 3.** AIoT Architecture with SDN and NFV.

Extending the 5G works to enhance SDN and NFV based IoT systems, Ref. [63] discussed a 5G management architecture that can provide distributed and on-demand deployment of network functions, provision of network slicing, flexible orchestration of network functions, and allocation of optimal workload [64]. The authors believe that the traditional mobile networks lack the flexibility and scalability required to offer the customized services of the current era. The authors in their work show how a 5G mobile network architecture based on the SDN and NFV based IoT systems can enhance the working of the entire model and produce the desired outcomes.

Authors in [65] present a Q-learning-based, QoS-aware robust resource management scheme for industrial IoT systems using a non-orthogonal multiple access scheme. The method is very effective in maintaining the diverse QoS requirements in future generations of IoT networks. Authors in [66] discuss a new approach of infusing SDN and NFV with IoT infrastructures by slicing end-to-end multiple networks segments and building an application that holds the capability to recover IoT services once out of order. In this model, applications are deployed as SDN controllers on top of the SDN network to bring about a high degree of interoperability, security, privacy, quality, flexibility, network slicing, energy efficiency, and virtualization [67,68]. Edge computing and AI can improve performance by reducing the bandwidth requirement in the backbone network.

IoT stemmed from a concept that prevailed in 2010 known as the Web of Things [69] where APIs such as REST and SOAP made interaction through interfaces feasible. It was in the same year that advancements began to make the transition from the Intranet of Things to the IoT [70].

## 5. Evolution of IoT Architecture

In this section, we shall traverse through the evolution of architectural models of IoT over the years and their advancements into the domains of SDN and NFV. The IoT concept was firstly introduced in 1982 when an Internet-connected cake machine was introduced at Carnegie Mellon University [71]. The term "IoT" was firstly coined by professor Kevin Ashton in the late 90s [72]. From 2000 to 2010, many electronics companies, such as LG, IBM, Bosh, etc., started working on the development of Internet-enabled electronic appliances. However, there has been a massive development in the field of IoT in the last decade.

The evolution of IoT devices with respect to time (from 2010–2021) is shown in Figure 4. Due to the massive availability of funding and the introduction of new IoT companies, the development of IoT devices has been boosted since 2010. The total number of IoT devices exceeded 6.2 billion (Bn) in 2010, and financial strength, government initiative, and increased focus of large electronics companies in this field were the main reasons for such a massive amount of IoT devices being developed [72,73].

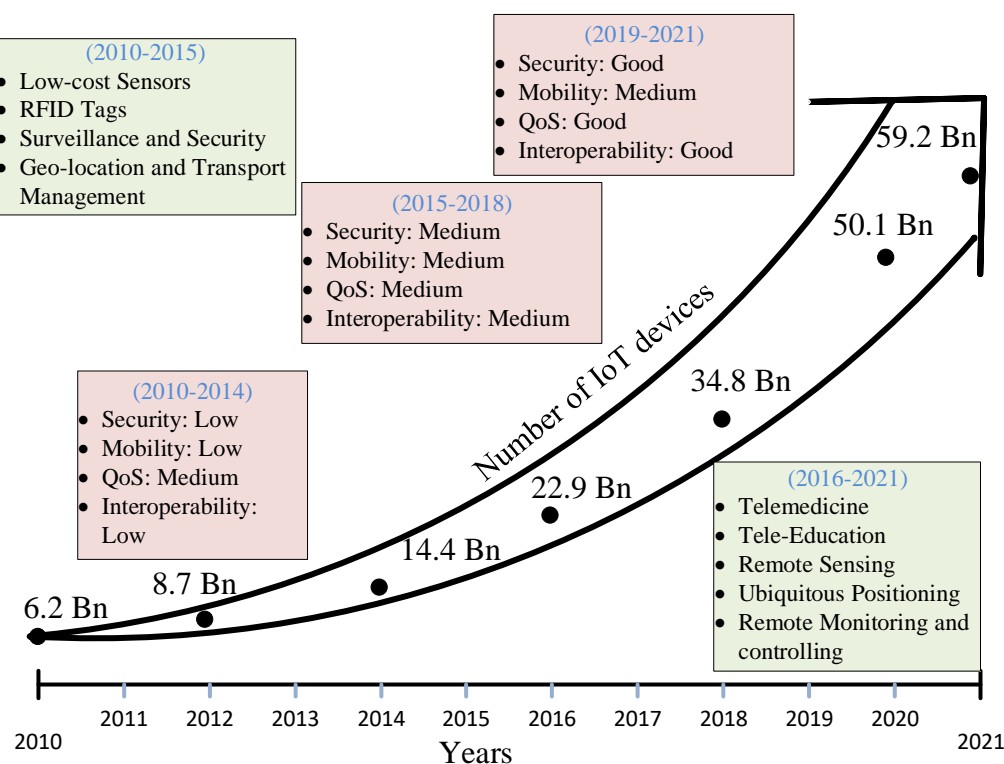

**Figure 4.** IoT Evolution with timeline.

The number of IoT devices reached 14.4 Bn in 2014. From 2010 to 2014, the development of radio frequency identification (RFID), low-cost sensors, surveillance and security devices, geo-location, and transportation management was a major focus of IoT companies. Security, mobility, inseparability, and QoS features were initially undervalued. However, these characteristics were improved over the next four years, with the number of IoT devices surpassing 34.8 Bn in 2018 [72]. Telemedicine, tele-education, remote sensing, ubiquitous positioning, remote monitoring, and controlling are all examples of IoT services that have been expanded. IoT companies' focus on improving security, mobility, and interoperability features in IoT devices has led to a global figure of 59.2 Bn IoT devices by 2021 [72]. In the following sub-sections, we explain the evolution of IoT networks in due course of time.

### 5.1. IoT Gateway

Hao Chen et al. in [8] analyze three network architectures that can work as IoT gateways for transmission of data and information between the sensing and the network layer. They are (1) Personal Area Network (PAN), (2) Vehicle Network, and (3) Home Network. However, standardization of these gateways has not yet been achieved. As IoT is in a very primitive stage at this time, the gateways only enable communication between the mentioned layers and lack advanced integration due to heterogeneous capabilities. The authors in [8] propose a reference IoT gateway that addresses the problems of multiple interfaces, protocol conversion, and manageability. The gateway acts as a proxy between different networks to address integration problems. It acts as a bridge for communication between the sensing and the network layer. However, the challenges of power optimization, cost efficiency, and protocol lightweight still stand large in the sensing domain.

Dong Chen et al. in [74] propose a four-layered architecture of IoT comprised of (1) Perception, (2) Heterogeneous Network Access, (3) Data Management, and (4) Intelligent Services Layer. This model aims to enhance the security of the system and the communication to strengthen the trust between heterogeneous entities. This is achieved through injecting security mechanisms and protocols into each of the layers of the four-layered IoT architecture. In [74] mechanisms to enforce security are discussed at each of the four-layers and their verification is justified. However, security is an evolving phenomenon that needs to be upgraded with the pace of the upgrades and the additions of complex structures to the IoT architecture.

### 5.2. Nanotechnology

An architecture based on a lightweight and open software middleware layer aiming at achieving a balanced combination of RFIDs and Smart Objects is proposed by Evangelos et al. in [75] directing towards a model for IoT which is generic and flexible in the hopes of optimizing the costs associated with RFID deployments as well as technical, social, and educational challenges. The model proposes the use of nanotechnology to address these issues by producing cheap, non-toxic, and disposable electronic devices. Interaction, awareness, and representation are salient features of smart objects making them complex entities. RFIDs are known for simplicity and scalability. A middleware for optimizing the use of both entities in the architecture hopes to resolve the aimed challenges of IoT. More research highlighting the use of smart and nanotechnology in IoT is presented by Miao Yun et al. in [76].

In [3], Zhihong Yang et al. present the three-layered traditional IoT architecture as defined by China Mobile. Several shortcomings and challenges of the basic IoT models are highlighted in this research which include power and cost concerns, remote administration, sensor quality, and peripherals adaptability and flexibility. RFIDs are the most convenient form of sensors to gather information in terms of installation, ease of use, scalability, and power and cost efficiency; hence, they are the most popular choice for a traditional IoT structure. However, they are faced with many challenges. Xiaolin et al. in [77] highlight some of the major challenges faced by RFID technology. According to [77], RFIDs are prone electromagnetic interference. Collisions could occur as a result of multiple transmissions across the same wireless channel. Anti-collision protocols such as Query Tree, Binary Tree, and Frame Slotted ALOHA exhibit efficiency lesser than 50.

### 5.3. Wireless Gateway

It is thus far a fact that traditional IoT lacks uniform standardization in communication protocols and sensing technologies, in addition to constraints on computational power, memory, and other resources. To address these issues, Soumya Kani et al. [78] propose an IoT architecture consisting of a Wireless Gateway (WG). This gateway aims at entertaining issues of heterogeneity, manageability, and bridging legacy intranet to endpoint networks. It consists of three layers: (1) a sensing layer, containing M2M devices and endpoints, (2) a gateway API layer, and (3) an application layer. The outstanding feature of this system is

that it is capable of interacting with non-smart devices over Modbus in addition to being highly scalable and generic to be compliant with future standards as well as providing an incremental methodology for adding new services.

### 5.4. SDN IoT with Fog Computing

Slavica Tomovic et al. in [79] state that, due to the exponential increase in data due to IoT, bigger data pipes to the data center are required, which is not economically feasible and also increases the latency for delay-sensitive applications. In addition, Fei Li et al. in [80] offer an IoT platform-as-a-service (PaaS) architecture based on WSO2 Stratos, an open-source PaaS solution as depicted in Figure 5. This technique is proposed on top of gateways to enhance the heterogeneity of the structure by mediating interfaces between gateways.

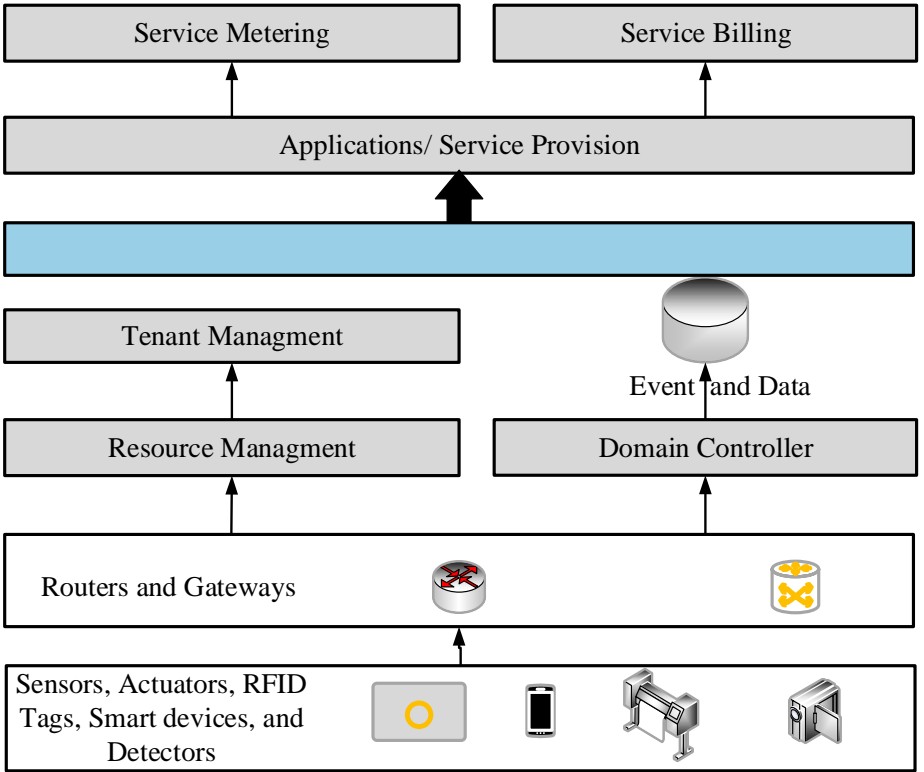

**Figure 5.** IoT PaaS Architecture.

The PaaS IoT optimizes the architecture's scalability and efficiency while providing multi-tenancy. However, further research is needed on system resource optimization and QoS. To counter this, they propose a distributed SDN architecture with geo-distributed fog nodes in an intermediate layer as depicted in Figure 6. The novelty of the system is that the fog orchestration is delegated to SDN controllers [79] to achieve higher efficiency, while the SDN scalability issue is relieved by delegating some controller tasks to fog nodes. This system provides a high degree of scalability, QoS, real-time delivery, and mobility. However, security and energy-efficient scheduling are not addressed explicitly.

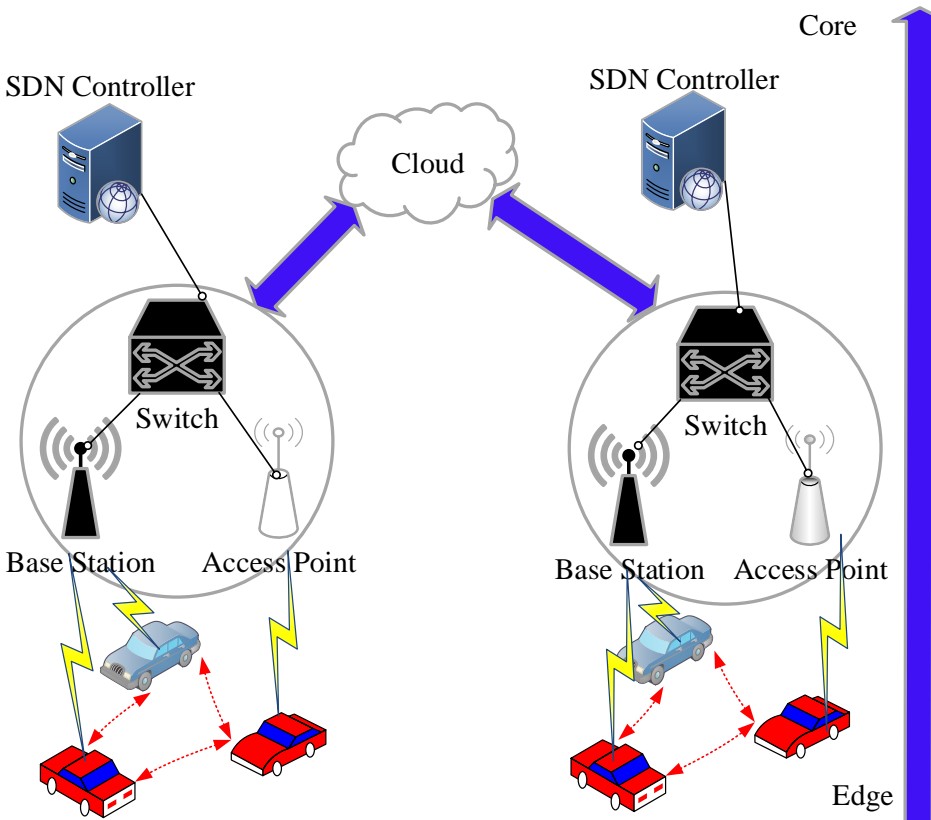

**Figure 6.** SDN Architecture of IoT based on Fog Computing.

### *5.5. UbiFlow*

Di Wu et al. proposed Ubiflow, a software-defined IoT system for ubiquitous flow control and mobility management in multi-networks. It builds on top of the MINA architecture [59] but has multiple controllers to divide an urban-scale SDN into different geographic partitions to achieve distributed control of IoT flows. Network controllers are distributed physically over geographic locations and are localized, meaning they have visibility to only their allowed domain. Scalability is enhanced in this approach but at a cost of overhead. No energy-efficient scheduling is offered and security is not focused [81].

### *5.6. Vertical and Horizontal IoT Models*

Currently, there are two different IoT models, vertical and horizontal IoT models [82]. In the vertical IoT model, the IoT node, gateway, and cloud are primarily managed by one single entity, achieving better compatibility among various elements. However, the vertical model has disadvantages as well, because the user is completely dependent on the vendor, and a different setup is required for each different task. This led to the development of the horizontal model, where multiple providers can work with the same underlying framework.

Yuhong Li et al. in [83] state that the focus of IoT applications has been to develop dedicated domains for all applications which resulted in data redundancy and problems in interoperability. It is difficult to introduce new services in the system and the changes are slow. The concept of horizontal IoT places all applications on a single domain which addresses these challenges profoundly. However, it decreases security and isolation in the system.

Jose L. Romero-Gaquuez et al. in [66] provide a solution to the main problems that can hamper Industrial 4.0 adoption for a medium high-factory (heterogeneity, management, congestion, and latency). They have used an open-source software solution architecture based on Open and Distance Learning (ODL) together with the IoT Data Management

(IoTDM) to orchestrate I4.0 infrastructure enabling interoperability and management of the diverse Industrial IoT (IIoT) devices. The authors have used SDN to enhance the programmability of the network. Scalability has been enhanced for the industrial IoT by this solution. The downsides are those important requirements of quality-of-service, energy-efficient scheduling, and security have not been adequately addressed.

### 5.7. Dist Block Net

P. K. Sharma et al. in [49] have proposed a distributed blockchains-based SDN architecture for IoT networks. In DistBlockNet architecture, all the distributed SDN controllers are interconnected in a distributed blockchain network manner. The main advantage is that the security issues in distributed SDN controller-based IoT architectures are addressed while providing the desirable feature of scalability. The beauty of this solution is that it provides threat prevention, data protection, access control detection of security threats, and mitigation of network attacks.

### 5.8. 5G Technologies

Comparisons of techniques for SDN and NFV in IoT are extended in Table 1 in a similar pattern. Justification for the comparison of various SDN and NFV solutions covered is given below in the same order. Interoperability is a critical challenge in IoT where multiple products of different vendors are connected. All solutions covered are partially compliant except the horizontal IoT theoretical model which by definition is compliant. As long as vertical IoT exists and applications remain domain-dependent, interoperability will remain a challenge. A secured environment is a fundamental requirement of IoT. Security, privacy, and trust become particularly challenging issues when things are connected to the global network.

**Table 1.** Traditional IoT Approaches.

| Approach | Technology | IOP | SP | Scal. | QoS | EES | Vir. | Het. | R | M | St. | MT |
|---|---|---|---|---|---|---|---|---|---|---|---|---|
| IOT Gateway | IoT | PC | PC | N | NA | NA | N | PC | NA | NA | NA | NA |
| Nanotechnology | IoT | PC | N | PC | NA | NA | N | PC | NA | NA | NA | NA |
| Task Computing Framework | IoT | PC | N | N | NA | NA | N | N | NA | NA | NA | NA |
| Tripple Based Distributed | IoT | PC | PC | N | NA | NA | N | N | NA | NA | NA | NA |
| UBIWARE | IoT | PC | N | NA | NA | NA | N | PC | NA | PC | N | NA |
| SOA | IoT | PC | PC | PC | NA | NA | N | PC | NA | NA | PC | NA |
| GSN | IoT | PC | N | PC | NA | NA | N | PC | NA | NA | NA | NA |
| Fosstrak | IoT | N | N | N | NA | NA | N | PC | NA | NA | PC | NA |
| TinyREST | IoT | PC | N | N | NA | NA | N | N | NA | NA | NA | NA |
| ROS | IoT | PC | PC | NA | NA | NA | N | PC | NA | NA | NA | NA |
| WSO2 Stratos | IoT/PaaS | PC | NA | PC | NA | NA | N | PC | NA | NA | NA | PC |
| Wireless Gateway (WG) | IoT | PC | NA | PC | NA | NA | N | PC | NA | NA | NA | NA |
| Smart Garbage Bin (SGB) | IoT | PC | PC | PC | NA | NA | N | N | PC | PC | NA | NA |

FC: Full Compliance; PC: Partial Compliance; N: No/Not Compliant; Y: Yes; and NA: Not Applicable.

Only two of the eight solutions discussed, "Full SDMN IoT with NFV" and "SDN IoT with Blockchain," can provide adequate security and privacy. IoT applications must be designed from the ground up to enable extensible services and operations. This means the ability to add new devices, services, and functions must be built-in without negatively affecting the quality of existing services. Out of the 10 solutions discussed, SDIoT is partially compliant, and scalability issues arise because it is hard to manage the large network in the case of a single SDSec logical element. An experimental evaluation is lacking.

Flow-based QoS scheduling ensures that the various QoS parameters and metrics at the application, network, and perception layers are optimum. It has been found that all solutions that used SDN in conjunction with NFV provided enhanced flow-based QoS

as they implement a heterogeneous cross-layer approach. Using fog computing with SDN also provides better flexibility to manage the QoS for various applications. Based on the literature review, it is found that IoT solutions that work on SDN in conjunction with NFV offer energy-efficient scheduling as they implement a heterogeneous cross-layer approach [84].

Figure 7 shows a software architecture for the 5G IoT network. There is data transfer to and from SD-BSs to the sensors, actuators, RFID tags, detectors, and other smart devices. This data is further transferred to the smart core network through SD-GWs [62]. Finally, different sectors such as smart grid, smart city, smart transportation are managed and monitored intelligently. These sectors can be further infused with blockchain and edge computing to enhance security, scalability, latency, and efficiency.

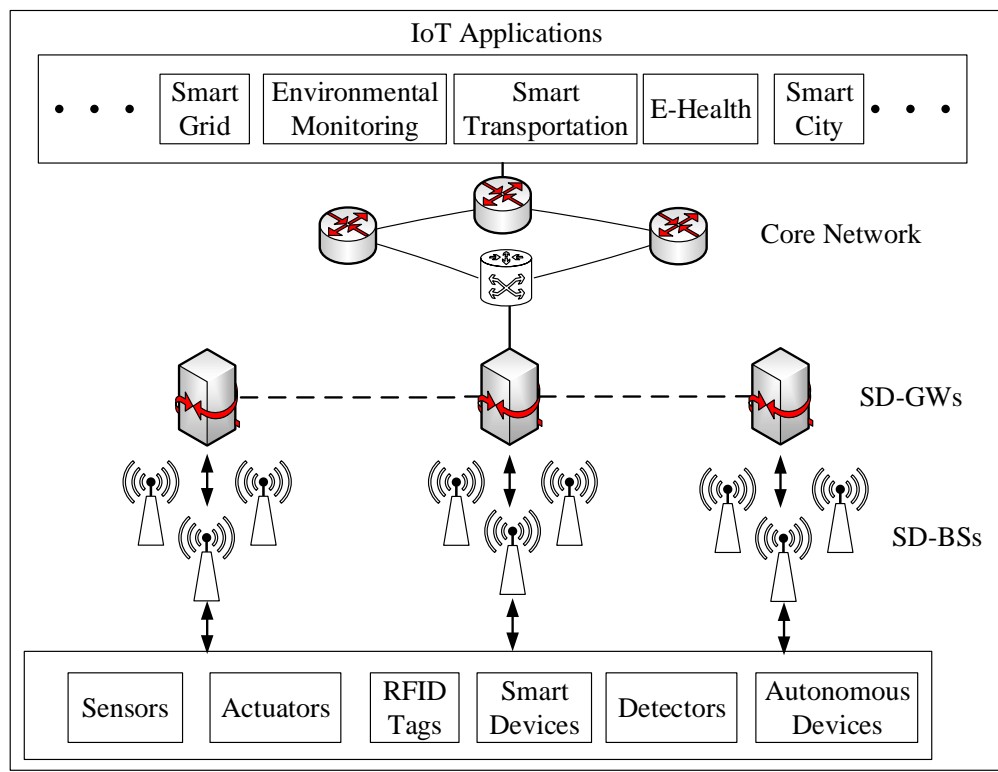

**Figure 7.** A Network Architecture for Software-Defined 5G-IoT.

The SDN-based, NFV infused IoT model requires to be injected with higher degrees of intelligence to enable the existing infrastructure and approaches to achieve better compliance and show higher performance and improved results. According to studies and surveys by Gartner, security is the number one concern of IT today and a key inhibitor in IoT and cloud adoption. Table 1 explicitly shows that security and privacy have not been the focus of the architectures of IoT proposed in this paper.

AI and Machine Learning (ML) based approaches should be exploited to implement Privacy Enhanced Techniques (PETs) and achieve the desired prioritization, anonymity security, and privacy. Fog computing and Blockchain technologies enhance security, but the intensive proof of concept computations add up to the latency and hence degrade performance. It is more suitable to use directed, acyclic graph distributed ledgers for IoT devices that do not require extensive computations. The need of the hour is to design and implement ways to integrate AI with the existing system to save cost and time, all the while avoiding major disruptions in the cycle.

## 6. State-of-the-Art IoT Simulators

New IoT frameworks, designs, and products need to be tested and verified before they are deployed or rolled out at a large scale into the production environment [85]. Creating a testing environment with a large number of IoT devices and a huge network may not always be a viable solution financially as well as technically [86]. As a proof-of-concept (PoC), IoT lifecycle [87] comprises of both physical and virtual elements; it is a best-suited solution to verify the virtual elements on a simulated environment and proceed to replicate the physical elements on a testbed through experimentation, before rolling out to the production. In this section, we will analyze some of the major simulators involved in the IoT lifecycle.

### 6.1. DPWSim

DPWSim is a full stack simulator [86] which uses the profile of the Service for Web Services (DPWS) standard [88]. This standard enables the use of resource-constraint devices which are an integral part of the IoT ecosystem. DPWSim offers secure messaging, dynamic discovery, service description, service invocation, and publish–subscribe eventing to prototype, develop, and test IoT elements.

### 6.2. iFogSim

iFogSim is another full stack simulator but, unlike DWPSim, it offers support for other enabling technologies and protocols as well, such as fog computing and Cloudsim Toolkit. This simulator is based on Cloudsim Toolkit [89]. iFogSim offers facilitation in the evaluation of end-to-end latency, network congestion, power usage, operational expenses, and QoS.

### 6.3. Cooja/Contiki

The practicality of Contiki is high and it is one of the very few simulators, such as OMNET++, to provide security with a custom extension. If we talk about the built-in IoT standard, it supports a large range of protocols that are all supported by the Contiki operating system [90]. It is especially generic and focuses on low power IoT devices or sensors. It provides dynamic loading and unloading of the applications which is open source, lightweight, and highly portable. It is also highly memory efficient and uses 2 KB of RAM and 40 Kb of ROM. It also provides security with a custom extension.

### 6.4. NS-3

NS-3 uses the perceptual layer of IoT to create realistic network simulations in WSN [91]. It is a successor of NS-2 and is still under evolutionary developments. It has native support for 6LoWPAN over 802.15.4 but it also lacks the support for application layer protocols. The code length is reduced in NS-3, which purely uses C++, as compared to NS-2 which uses OTCL and has a greater code length. Python script support is also enabled in NS-3.

### 6.5. QualNet

QualNet is one of the first commercial [92] simulators which falls under the category of Network Simulator [86] used to target simulations for network solutions. QualNet provides a comprehensive, versatile, and user-friendly GUI for creating and simulating networks.

### 6.6. Netsim

Netsim is another network simulator which is a hybrid software and hardware simulation system for performing real-time realistic operations [93]. Netsim simulates network traffic through the usage of digital packet generation to path packets and transfer frames. Netsim allows the building of whole labs with custom network simulations of hundreds of devices in the same topology.

### 6.7. OMNet++

Another popular simulator that is extensively used in WSN and IoT research is OM-Net++ [94]. It is extensible and can incorporate urban mobility tools for simulating urban mobility. INIT is an open-source library and provides different protocols as well as agents for the OMNet++ simulations environment. INIT, along with other mobility frameworks, is integrated to develop the simulation environment in OMNeT++ [95]. Specialized vehicular network simulation tools can be used with OMNeT++ to simulate the vehicular environment. It is widely used to simulate different smart traffic scenarios. The output of OMNeT++ can be simulated with some other famous network simulation environment, i.e., Matlab. It has both a commercial and academic version available. OMNet++ lacks the support of application layer protocols and radio models which are specific to IoT.

### 6.8. GridSim

Gridsim toolkit is used for grid computing distributed computing which provides a comprehensive utility to simulate different classes of heterogeneous applications, users, resources, resource brokers, and schedulers [96]. Each user has a private resource broker, hence it can be used to optimize the requirements and objectives of the owner. All users need to submit their jobs to the central scheduler which can be used to perform further global optimization.

### 6.9. Mininet-WiFi

Mininet-WiFi is another tool that simulates network environments for OpenFlow and software-defined scenarios with high-fidelity. Mininet-WiFi is an extension of Mininet to support WiFi by adding virtualized WiFi stations and access points based on the most common Linux wireless device driver [97].

*Comparative Analysis:* Based on the extensive literature review, IoT key parameters have been graded in terms of compliance and whether a parameter is supported. We used numerous parameters in Tables 1 and 2, but the parameters (interoperability, security and privacy, scalability, flow-based QoS, energy-efficient scheduling, mobility, and reliability) are the most important challenges that IoT is facing in the widespread deployment and realization of the true potential to humanity.

**Table 2.** SDN/NFV Based IoT Approaches.

| Approach | Technology | IOP | SP | Scal | QoS | EES | Vir. | Het. | DRU | SI | NS | MT | ACD | AI |
|---|---|---|---|---|---|---|---|---|---|---|---|---|---|---|
| MINA | SDN | PC | PC | PC | PC | NA | N | FC | PC | NA | N | N | N | N |
| SDIoT | SDN | PC | PC | PC | NA | NA | Y | FC | PC | NA | N | N | N | N |
| UbiFlow | SDN | PC | PC | FC | PC | PC | N | FC | FC | Y | N | N | N | N |
| Horizontal IoT | SDN | FC | PC | FC | NA | N | N | FC | FC | NA | N | N | N | N |
| IIoT SDN | SDN | PC | PC | FC | NA | NA | N | PC | PC | NA | N | N | N | N |
| SDN IoT with FOG | SDN/FOG | PC | PC | FC | FC | N | N | FC | FC | NA | N | N | N | N |
| DiskBlockNet | SDN/Blockchain | PC | FC | FC | NA | N | N | PC | NA | NA | N | N | N | N |
| SDN IoT with NFV | SDN/NFV | PC | PC | FC | FC | FC | Y | FC | FC | Y | NA | NA | N | N |
| Softair | SDN/NFV | PC | PC | FC | FC | FC | Y | FC | FC | Y | Y | Y | N | N |
| Full SDMN with NFV | SDN/NFV | PC | FC | FC | FC | FC | Y | FC | FC | NA | Y | Y | N | N |

FC: Full Compliance; PC: Partial Compliance; N: No/Not Compliant; Y: Yes; NA: Not Applicable; IOP: Interoperability; SP: Security and Privacy; Scal: Scalability; QoS: Quality of Service; EES: Energy Efficient Scheduling; Vir: Virtualization; Het: Heterogeneity; R: Reliability; M: Mobility; St: Standardization; MT: Multi Tenancy; DRU: DRU; SI: Subnet Isolation; NS: Network Slicing; ACD: Autonomous Computing Device; and AI: AI.

Table 2 consolidates the approaches of traditional IoT architectures against the key parameters for comparison. The earliest types of IoT were having relatively simple protocols with the goal of enabling interconnection between devices. Traditional IoT technology was incapable of addressing the key parameters of IoT, and there was no emphasis on

optimising architectures at the time. As a result, these issues were identified as aspects of IoT challenges and future development [49,70].

Therefore, it can be observed that the approaches discussed play their part to ensure interoperability and heterogeneity only as their salient features and fail to cover the rest of the advanced domains. Some solutions propose gateways [8] and others present service-oriented architectures (SOA) [53] in addition to robotics-based structures [53] to extend their coverage towards security and privacy of data.

A few mentioned techniques providing ease of scalability are nanotechnology, SOA, wireless gateways, PaaS IoT, and Global Sensor Networks [53,75,78,80]. UBIWARE [53] and architectures for Smart Waste Management offer some degrees of Mobility, and Fosstrak and SOA provide certain degrees of Standardization [53]. Only PaaS IoT offers multi-tenancy [80].

## 7. IoT Challenges and Future Directions

The frameworks of IoT face several challenges that need to be addressed for a successfully integrated world. Some salient challenges IoT must face are discussed in the sections that follow.

### 7.1. Mobility and Dynamic Environment

It is challenging for IoT devices to communicate in a dynamic environment, where devices continuously change their positions. People, devices, and vehicles are carrying those IoT sensors or devices and communicating in the real world. This has shifted the traditional static networks to highly dynamic networks, where mobility is an important factor, which was not in the case of traditional computer networks. In IoV, the network topology is continuously changing and vehicles are communicating with each other to enhance passenger safety. This type of dynamic communication has brought a revolution in the field of intelligent transport systems [98,99].

IoT must support connecting applications and devices to the requisite services for the mobile users [11]. A service provider often finds itself connecting to devices or providing service to nodes that are constantly on the move or to mobile edge devices. A mobile user may face service disruption when the device switches from one zone to another or a handover takes place between gateways. This disruption of service reduces the QoS and degrades the system as a whole. An IoT framework demands the architecture to be flexible, scalable, and efficient enough to provide uninterrupted service to mobile devices on the network. Therefore, developing some robust and efficient algorithms for mobility management and coping with a dynamic environment is essential. In IoT networks, machine learning can help predict the dynamically changing environment.

### 7.2. Security and Privacy

The more devices we bring online, the higher becomes the need to secure the data being transmitted between machines and users. As we expand our mesh of inter-connectivity, our data become more prone to cyber-attacks and crimes. With this advancement comes the greater responsibility of securely transmitting and sharing our data and making communication private [100]. Some new techniques, such as blockchain-based security enhancement in IoT networks, can be used to improve security and privacy. Similarly, more robust algorithms can be designed to improve IoT network security and privacy.

### 7.3. Interoperability

IoT does not differentiate between the devices on the network, which means that it is a mesh of heterogeneous devices and different operating systems. It is a challenge to manage such vast heterogeneity and enable devices of different domains and contexts to communicate and interact successfully with each other [101]. Therefore, it is essential to concentrate on architectures and protocol developments that can accommodate the seamless coexistence of multiple radio access technologies in the IoT environment in order to meet

the QoS requirements of IoT devices. In this area, machine learning and blockchain-based technologies can play a significant role.

### 7.4. Reliability

Some applications of IoT thrive on real-time communication and accurate information. Taking smart health as an example to explain the significance of reliability, the system can only successfully operate if the emergency is intimidated (1) in time and (2) with an accurate and complete description of the incident [11]. In security and privacy-sensitive IoT networks such as healthcare IoT, more advanced ML techniques, such as federated learning, can be implemented [102]. Similarly, a blockchain-based IoT network architecture can improve the IoT network's security and reliability.

### 7.5. Power Consumption

The devices on the IoT need to be energy efficient, as it is not always possible to have each device connected to power sources. As most of the nodes function on batteries and in remote, far-off locations, they need to be very smart in consuming energy. Additionally, we need to transform our IoT nodes to be greener and environmentally friendly so that they are not only efficient in consuming energy but also have a minimum impact on the environment [103,104]. As a matter of fact, it is critical to concentrate on energy-efficient and battery-saving techniques as well as algorithms for IoT networks. In addition, the hardware architecture of power-hungry IoT devices could be redesigned to save energy while maintaining QoS requirements.

## 8. Conclusions and Future Work

The conventional architectures and approaches of IoT do not have the tendencies to match the new age requirements. In this paper, we have comprehensively put forward the concepts of traditional IoT, its drawbacks, and how they can be addressed using SDN-based smart IoT networks. Applications and challenges of IoT are discussed along with the architecture of traditional IoT, leading to the motivation behind SDN-based IoT with disruptive technologies such as NFV, fog computing, and blockchain infused into the model. Furthermore, multiple solutions of SDN-based IoT networks are critically analyzed and the strengths and drawbacks of each are consolidated in a table against the necessities of IoT, as per their degree of compliance.

Research on the multiple solutions of SDN/NFV based IoT approaches can be extended towards achieving, enhancing, or optimizing the IoT key parameter and requirements of (1) Interoperability (2) Power Management (3) Security (4) QoS per-flow, and (5) Service Reliability. There is a lot of room for improvement along with these parameters to produce better, smarter, and more efficient solutions utilizing cutting-edge technologies such as autonomous device computing and AI. The domain of power management can be enriched by exploring the untapped areas of the Internet of sustainable battery-less things. We tend to take this work forward and propose a Distributed SDN-based Smart IoT architecture that addresses the limitations of a centralized SDN Model.

**Author Contributions:** M.A.A., N.A., M.W.A. and B.S. performed formal analysis and original draft preparation. M.A.K. and A.M. proposed the main ideas and validated analysis. M.W.A., A.W.M., A.M. and J.A. crystallized framework and also revised the manuscript. All authors have read and agreed to the published version of the manuscript.

**Funding:** This research received no external funding.

**Informed Consent Statement:** Not applicable.

**Data Availability Statement:** Not applicable.

**Conflicts of Interest:** The authors declare no conflict of interest.

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
