# Peer review of "Evolution towards Smart and Software-Defined Internet of Things"

_ai, doi:10.3390/ai3010007_

Round 1

Reviewer 1 Report

This is a review paper on the Internet of Things. The author introduced IoT classification, smart IoT architecture, smart software-defined IoT, evolution of IoT architecture, and IoT simulators. However, there are the following problems.

  1. The author introduced a lot of content, but it was too straightforward and ignored the connection between the works. In other words, the author did not provide a reasonable summary of the various stages of IoT development.
  2. The summaries in some research areas were not very sufficient, such as “environmental IoT”, “application layer”.
  3. In the motivation section, the author mentioned a missing link in terms of the evolution of the IoT architectures starting from primitive and traditional frameworks to highly evolved software-defined and virtualized platforms. However, the paper did not elaborate on how to solve this problem.
  4. The paper lacked prospects for the future and did not point out future research points.

Author Response

Responses to Comments from the Editor and Reviewers

The authors have thoroughly revised the paper reflecting all the constructive comments of the editor and the reviewers. The comments were helpful in improving the quality of our paper. We have clarified all the issues mentioned by the reviewers, and the manuscript has been revised based on all the comments from the reviewers. The detailed response to each review comment is provided below each comment for reviewers to easily check it.

Reviewer 2 Report

This work discusses the conventional IoT networks and the need for Software-Defined Networking (SDN) and Network Function Virtualization (NFV), followed by an analysis of the SDN and NFV solutions to implement IoT in various possible ways.

This study is very interesting but not completely technically sound. The section about IoT Classification and Applications needs to be improved. In particular, the authors should extend section 2.2, a crucial aspect of IoT scenarios.

The paper is well organized.

Author Response

(The authors gave the same response as above.)

Reviewer 3 Report

Thank you for inviting me to review this work. It was a pleasure to read and try to understand the message of the authors and the scientific contribution of this work. The article provides comprehensively the concepts of traditional IoT, its drawbacks, and how they can be addressed using SDN-based smart IoT networks. Applications and challenges of IoT are discussed along with the architecture of traditional IoT, leading to the motivation behind SDN-based IoT with disruptive technologies such as NFV, fog computing, and blockchain, etc. The paper is more a state of the art review, however a very interesting and thorough one, making it an important contribution for future research in the addressed field.

Author Response

(The authors gave the same response as above.)

Round 2

Reviewer 1 Report

Authors have improved the paper accordingly. However, the following references (with certain topics) should be added to the content and discussed in where applicable.

[1]  DTaxa: An Actor-Critic for Automatic Taxonomy Induction. Engineering Applications of Artificial Intelligence

[2]Energy saving of buildings for reducing carbon dioxide emissions using novel dendrite net integrated adaptive mean square gradient. 

Author Response

Dear Reviewer:

Authors have improved the paper accordingly. However, the following references (with certain topics) should be added to the content and discussed in where applicable.

[1]  DTaxa: An Actor-Critic for Automatic Taxonomy Induction. Engineering Applications of Artificial Intelligence

[2]Energy saving of buildings for reducing carbon dioxide emissions using novel dendrite net integrated adaptive mean square gradient. 

Responses to the Reviewer's Comments

Thank you for your feedback and review. We have added these related studies in the paper. Please see Ref 104 and 106. Additionally, discussion can be seen in lines 754-756 and 759-764.
